# Polymorphisms of *ATP-Binding Cassette, Sub-Family A, Member 4* (rs560426 and rs481931) and Non-Syndromic Cleft Lip/Palate: A Meta-Analysis

**DOI:** 10.3390/life11010058

**Published:** 2021-01-15

**Authors:** Mohammad Moslem Imani, Masoud Sadeghi, Santosh Kumar Tadakamadla, Annette Brühl, Dena Sadeghi Bahmani, Mohammad Taheri, Serge Brand

**Affiliations:** 1Department of Orthodontics, School of Dentistry, Kermanshah University of Medical Sciences, Kermanshah 6713954658, Iran; mmoslem.imani@yahoo.com or; 2Medical Biology Research Center, Kermanshah University of Medical Sciences, Kermanshah 6714415185, Iran; sadeghi_mbrc@yahoo.com or; 3Students Research Committee, Kermanshah University of Medical Sciences, Kermanshah 6715847141, Iran; 4Senior Research and National Health and Medical Research Council Fellow, School of Dentistry and Oral Health, Griffith University, Gold Coast 4214, Australia; santoshkumar.tadakamadla@griffithuni.edu.au; 5Center for Affective, Stress and Sleep Disorders (ZASS), Psychiatric University Hospital Basel, 4002 Basel, Switzerland; annette.bruehl@upk.ch (A.B.); dena.sadeghibahmani@upk.ch (D.S.B.); 6Department of Clinical Research, University of Basel, 4031 Basel, Switzerland; 7Sleep Disorders Research Center, Kermanshah University of Medical Sciences, Kermanshah 6719851115, Iran; 8Substance Abuse Prevention Research Center, Kermanshah University of Medical Sciences, Kermanshah 6715847141, Iran; 9Departments of Physical Therapy, University of Alabama at Birmingham, Birmingham, AL 35209, USA; 10Urogenital Stem Cell Research Center, Shahid Beheshti University of Medical Sciences, Tehran 1983963113, Iran; mohammad.taheri@sbmu.ac.ir; 11Department of Sport, Exercise and Health, Division of Sport Science and Psychosocial Health, University of Basel, 4052 Basel, Switzerland; 12School of Medicine, Tehran University of Medical Sciences, Tehran 1416753955, Iran

**Keywords:** ABCA4, polymorphism, variation, non-syndromic cleft lip/palate, meta-analysis

## Abstract

Background: A number of genes are associated with the incidence of non-syndromic cleft lip/palate (NSCL/P). Studies have shown a significant association between polymorphisms of ATP-binding cassette, sub-family A, member 4 (ABCA4) with the risk of NSCL/P. The present meta-analysis assessed the association between ABCA4 polymorphisms (rs560426 and rs481931) and the NSCL/P risk by reviewing case-control studies. Methods: Four databases (Scopus; Cochrane Library; Web of Science; and PubMed) were searched for articles published up to June 2020. The Review Manager 5.3 software was used to calculate the crude odds ratio (OR) and 95% confidence interval (CI). Both subgroup analyses for ethnicity and source of controls and a meta-regression related to publication year were conducted. Results: Of 94 retrieved studies, 12 were analyzed in this meta-analysis (2859 NSCL/P patients and 3792 controls for ABCA4 rs560426 polymorphism and 1333 NSCL/P patients and 1884 controls for ABCA4 rs481931 polymorphism). Overall, there was no significant association between both polymorphisms and the risk of NSCL/P. However, subgroup analysis demonstrated that there was a higher risk of NSCL/P for specific models: the allelic model (OR = 1.13; *p* = 0.03), the homozygote model (OR = 1.53; *p* = 0.04), and the recessive model (OR = 1.30; *p* = 0.03) in the Asian ethnicity for the rs560426 polymorphism. Conclusion: The findings confirmed that the NSCL/P risk was significantly associated with the G allele and GG genotype of rs560426 polymorphism but not for rs481931 polymorphism. There were no associations between both polymorphisms (rs560426 and rs481931) and the NSCL/P risk in those of European descent and the mixed ethnicities.

## 1. Introduction

More than 70% of all cleft lips/palates (CL/Ps) present without further anomalies and are thus nonsyndromic [1]. Its prevalence ranges from 1/700 to 1/1000, depending on the geographical area and ethnicity [2]. Non-syndromic cleft lip/palate (NSCL/P) has a multifactorial etiology and therefore both environmental and genetic risk factors can affect its occurrence. However, the link between different environmental factors and the disease is contradictory [1,3], except for maternal smoking. Maternal smoking, alcohol consumption, folic acid and vitamin deficiencies especially during the first trimester of pregnancy have been reported to increase the incidence of NSCL/P [4]. In addition, differences in craniofacial characteristics can depend on demographic factors such as age [5,6] and gender [5]. Considering the complex etiology of NSCL/P, studies have shown that gene–gene and gene–environment interactions may be related to NSCL/P susceptibility [3,7,8]. Subsequent genome-wide association studies (GWAS) showed that a number of gene loci are closely associated with the incidence of NSCL/P, which was increased to 40 risk loci by identifying 14 novel loci in the Chinese population added to the 26 previously known risk loci [9,10,11,12]. In addition, an overlap between the genetics of nsCL/P and biologically relevant facial phenotypes was reported, suggesting that a decreased philtrum width can be caused by genetic risk SNPs of nsCL/P [13].

ATP-binding cassette, sub-family A, member 4 (ABCA4) belongs to the transmembrane protein superfamily [14], which is located on chromosome 1p22.1 [15,16]. The ABCA4 is an ATP-binding cassette transporter that is particularly expressed in the rod and cone photoreceptor cells of the vertebrate retina [14,15,16]. It is also expressed in the murine brain [17]. A systematic review reported the significant presence of ocular abnormalities in patients with NSCL/P [18]. Further, GWAS studies [19,20,21] confirmed an association between polymorphisms of ABCA4 and the NSCL/P risk, where one study [19] showed genome-wide significances of 8.14 × 10^−8^ and 5.01 × 10^−12^ for the rs560426 polymorphism in combined European and Asian cases and another study [20] reported 1.06 × 10^−12^ for the rs481931 polymorphism in Asian families. A third study [21], a genome-wide meta-analysis, showed a genome-wide significance of 3.14 × 10^−12^ for rs560426 in an Asian population. In addition, some studies showed a significant association between rs560426 and rs481931 polymorphisms of ABCA4 with the risk of NSCL/P [22]; in contrast, other studies found no significant association [8,23]. Given these contradictory results, it appears that the genetic basis of oral clefts is still unclear, and identification of additional risk factors for NSCL/P might greatly help in genetic counseling and may prevent the occurrence or further development of this condition [23]. There are no previous meta-analyses reporting the association of ABCA4 (rs560426 and rs481931) and the risk of NSCL/P. Therefore, the present meta-analysis aimed to evaluate the association between the most common polymorphisms of ABCA4 (rs560426 and rs481931) and the risk of NSCL/P in case-control studies.

## 2. Materials and Methods

This report conforms to the Preferred Reporting Items for Systematic reviews and Meta-Analyses (PRISMA) guidelines [24].

### 2.1. Literature Search Strategy

Four electronic databases (Web of Science; Cochrane Library; Scopus; and PubMed) were searched for articles published until 21 June 2020, without restrictions of language or publication date. The searched terms were: (“*ABCA4*” or “ATP-binding cassette subfamily A member 4” or “rs560426” or “rs481931”) and (“cleft” or “cleft lip” or “cleft palate” or “orofacial cleft” or “oral cleft”). In addition, we checked the references of retrieved articles, review articles, and GWAS studies to find potential articles.

### 2.2. Eligibility Criteria

One reviewer (M.S.) retrieved the studies from the databases, excluded the duplicate and irrelevant studies after reviewing the titles and abstracts. This was followed by full-text review of the eligible articles. The studies were included if they met the following inclusion criteria: (I) case-control design; (II) NSCL/P was the outcome of interest; (III) reporting *ABCA4* rs560426 (A > G) and/or rs481931 (C > A) polymorphisms, and (IV) having the required data to calculate the odds ratios (ORs) with 95% confidence intervals (CIs). The studies were excluded if they did not have the required data regarding genotype distributions or were animal studies, review articles, letters to the editor reporting previous studies, and family-based studies. The second reviewer (M.M.I.) checked the relevant articles based on the eligibility criteria. The differences between the two reviewers were resolved by the third reviewer (S.K.T.).

### 2.3. Data Extraction

Two reviewers (M.M.I. and M.S.) independently extracted the data from each study. The information retrieved from the studies included: the first author’s name, the publication year, the ethnic group, the source of controls, the mean age, the number of males in the two groups, the number of patients and controls with each genotype, the genotyping method, and the *p*-value of the Hardy–Weinberg equilibrium in controls. If there was a disagreement between the reviewers, the problem was solved by a third reviewer (S.B.).

### 2.4. Quality of Assessment

One reviewer (M.M.I.) rated the quality of each included article using the Newcastle–Ottawa Quality Assessment Scale (NOS). The NOS involves assessment of each article for quality in three components: selection, comparability, and exposure. A case-control study could have a minimum and maximum total score of 0 and 9, respectively; a higher score demonstrates better quality [25].

### 2.5. Statistical Analyses

An analysis was performed by the Review Manager 5.3 using crude OR and 95% CI to show the association between *ABCA4* polymorphisms and the risk of NSCL/P in the five genetic models [26]. The Z test was applied to evaluate the pooled OR significance. Heterogeneity across the studies was checked by the Cochrane Q test and I^2^ statistic. Heterogeneity was considered to be statistically significant if *p* < 0.1 or I^2^ > 50%. If there was no significant heterogeneity, the fixed-effects model (Mantel–Haenszel method) was used to estimate the values. Otherwise, we used the random-effects model (DerSimonian and Laird method). The Chi-square test was used to calculate the Hardy–Weinberg equilibrium in the control group of each study. Subgroup analysis was done according to the ethnicity and the source of controls. Meta-regression is a quantitative method used in meta-analysis to estimate the impact of moderator variables on the study effect size. The Comprehensive Meta-Analysis 2.0 was used to derive a funnel plot using the Egger’s and Begg’s tests and *p* < 0.05 indicated significant existence of publication bias. To evaluate the stability of the results, the following sensitivity analyses were applied: “cumulative analysis” and “one study removed”. The analysis was checked by three reviewers in a discussion (S.B., D.S.B., and M.S.).

## 3. Results

### 3.1. Study Selection

Among the 94 retrieved studies, after removing duplicate (61 studies) and excluding the book chapters, conference papers, and studies without reporting the association between the *ABCA4* polymorphisms and NSCL/P risk or irrelevant studies (11 studies), a total of 23 studies were evaluated (Figure 1). Of 23 studies, eleven were excluded due to the following reasons: four had no control groups, four did not have the required data, two were letters to the editor reporting previous studies, and one evaluated the mothers of children. Finally, 12 studies were included in this meta-analysis.

### 3.2. Study Characteristics

Table 1 shows some features of the studies included in this meta-analysis. The studies were published between 2011 and 2018. Five studies [8,27,28,29,30] reported the risk of NSCL/P related to *ABCA4* polymorphisms in the Asian ethnicity, four [7,9,23,31] in the mixed ethnicity, and three [32,33,34] in the European descent ethnicity. In six studies [23,27,29,30,32,33], the source of controls were hospitals, while for six other studies [7,8,9,29,31,34], the source of controls was the general population. All included studies reported the *ABCA4* rs560426 polymorphism, but only six [7,8,30,32,33,34] reported the *ABCA4* rs481931 polymorphism. There were different genotyping methods among the studies presented in Table 1. The Hardy–Weinberg equilibrium was not seen in controls of two studies [8,27] reporting the *ABCA4* rs560426 polymorphism and two studies [8,33] reporting the *ABCA4* rs481931 polymorphism. We included these studies in the analyses because by excluding them, the number of studies would have been reduced, while this could have been one important reason for an overall biased pattern of results. There were 2859 NSCL/P cases and 3792 controls in the meta-analysis of the *ABCA4* rs560426 polymorphism and 1333 NSCL/P cases and 1884 controls in the meta-analysis of the *ABCA4* rs481931 polymorphism.

### 3.3. Quality Assessment

All studies were scored for quality (Table 2), and all had a score of ≥7.

### 3.4. Pooled Analysis

Table 3 shows the pooled results (combining data from each individual study) for the risk of NSCL/P in those with the *ABCA4* rs560426 polymorphism from the twelve studies. The heterogeneity in allelic, homozygote, heterozygote, and dominant models was high (I^2^ > 50%), and therefore the random-effects model was used. In the recessive model, the fixed-effects model was used because of low heterogeneity (I^2^ < 50%). The pooled OR was 1.01 (95% CI: 0.88, 1.15; *p* = 0.92; I^2^ = 72% (*P*_h_ or *P*_heterogeneity_ < 0.0001)) in the allelic model (G vs. A), 1.08 (95% CI: 0.79, 1.47; *p* = 0.64; I^2^ = 77% (*P*_h_ < 0.00001)) in the homozygote model (GG vs. AA), 0.93 (95% CI: 0.73, 1.17; *p* = 0.53; I^2^ = 76% (*P*_h_ < 0.00001)) in the heterozygote model (AG vs. AA), 0.89 (95% CI: 0.70, 1.14; *p* = 0.37; I^2^ = 81% (*P*_h_ < 0.00001)) in the dominant model (AG + GG vs. AA), and 1.08 (95% CI: 0.91, 1.26; *p* = 0.38; I^2^ = 36% (*P*_h_ = 0.10)) in the recessive model (GG vs. AA + AG). Overall, individuals with the *ABCA4* rs560426 polymorphism were not at higher risk for NSCL/P than those without the *ABCA4* rs560426 polymorphism.

The pooled results for the risk of NSCL/P related to the *ABCA4* rs481931 polymorphism based on the findings from the six studies are shown in Table 4. The heterogeneity in all analyses was low, and therefore, fixed-effect models were used (I^2^ < 50%). The pooled OR was 0.92 (95% CI: 0.83, 1.02; *p* = 0.12; I^2^ = 35% (*P*_h_ = 0.17)) in the allelic model (A vs. C), 0.85 (95%CI: 0.68, 1.05; *p* = 0.13; I^2^ = 33% (*P*_h_ = 0.19)) in the homozygote model (AA vs. CC), 0.88 (95%CI: 0.75, 1.03; *p* = 0.12; I^2^ = 16% (*P*_h_ = 0.31)) in the heterozygote model (CA vs. CC), 0.87 (95%CI: 0.75, 1.00; *p* = 0.06; I^2^ = 29% (*P*_h_ = 0.22)) in the dominant model (CA + AA vs. CC), and 0.89 (95%CI: 0.74, 1.09; *p* = 0.26; I^2^ = 22% (*P*_h_ = 0.27)) in the recessive model (AA vs. CC + CA). Based on the genetic models, there was no significant risk of NSCL/P in those with the *ABCA4* rs481931 polymorphism.

### 3.5. Subgroup Analysis

The first subgroup analysis evaluated the effect of ethnicity and the source of controls in the association of NSCL/P with the *ABCA4* rs560426 polymorphism (Table 5). A significant risk for NSCL/P was observed in the Asian ethnicity for the allelic model (OR = 1.13; 95% CI: 1.01, 1.27; *p* = 0.03), the homozygote model (OR = 1.53; 95% CI: 1.01, 2.31; *p* = 0.04), and the recessive model (OR = 1.30; 95% CI: 1.03, 1.63; *p* = 0.03). The results did not demonstrate a risk of NSCL/P related to the rs560426 polymorphism among people of European descent and mixed ethnicities (such as Brazilian and Mexican ethnicities). Furthermore, the source of controls did not seem to have any effect on the association of NSCL/P and the *ABCA4* rs560426 polymorphism.

The second subgroup analysis assessed the risk of NSCL/P related to the *ABCA4* rs481931 polymorphism according to the ethnicity and the source of controls (Table 6). There was no association between the risk of NSCL/P and the rs481931 polymorphism based on ethnicity or source of controls.

### 3.6. Meta-Regression

The results of the meta-regression for finding the effect of the publication year and the number of participants on the results showed that publication year was not a significant confounding on the association between both rs560426 and rs481931 polymorphisms and the risk of NSCL/P (Table 7). In addition, the number of participants was not a significant confounding factor on the association between rs560426 polymorphism and the risk of NSCL/P, but the number of participants was a significant confounding factor for the rs481931 polymorphism on the pooled ORs in allele, homozygote, and dominant models.

### 3.7. Sensitivity Analysis

The sensitivity analyses (“cumulative analysis” and “one study removed” analyses), identified that the pooled ORs under all genetic models were stable and trustworthy (the values of the pooled ORs did not change). Further, excluding two studies [8,28] with *p*-values < 0.05 for the Hardy–Weinberg equilibrium in controls did not change the overall result regarding the risk of NSCL/P in relation to the *ABCA4* rs560426 polymorphism. Excluding two other studies [8,33] did not change the overall result regarding the risk of NSCL/P with the *ABCA4* rs481931 polymorphism (not presented in tables). The quality score of all studies was high (≥7) and therefore we could not use a sensitivity analysis (excluding the studies with low quality) for this topic.

### 3.8. Publication Bias

Figure 2 shows the funnel plot of the association between *ABCA4* polymorphisms and the risk of NSCL/P using the five genetic models. The Egger’s and Begg’s tests did not reveal any publication bias (*p* > 0.05) except for four states. Both tests for allele and homozygote models revealed a publication bias and Egger’s test for heterozygote and recessive models revealed a publication bias (*p* < 0.05).

## 4. Discussion

The present meta-analysis evaluated the associations between rs560426 and rs481931 polymorphisms of *ABCA4* and the risk of NSCL/P based on five genetic models. The key findings were as follows: While there was no significant association between both polymorphisms and the risk of NSCL/P, in contrast, subgroup analyses demonstrated that there was a higher risk of NSCL/P for the allelic model, the homozygote model, and the recessive model in the Asian ethnicity for the rs560426 polymorphism. Further, there were no associations between both polymorphisms (rs560426 and rs481931) and the NSCL/P risk in the European descent and the mixed ethnicities. The present results showed association of the rs560426 polymorphism and the risk of NSCL/P among the Asian, but not among the European, descent. Further, the meta-regression showed that the number of participants was a confounding factor for the association between the rs481931 polymorphism and the risk of NSCL/P. Overall, the ethnicity and the number of participants may act as significant confounders in this association.

NSCL/P is a complex congenital anomaly that shows both clinical and genetic heterogeneity and the genetic basis of NSCL/P has remained unclear [23].

Further, there is little biological evidence to support the role of *ABCA4* in facial cranial morphogenesis, especially since Abca4 null mice do not show cleft palate [35]. It turned out that the rs560426 G allele showed a 1.36- [30] and 1.34-fold [31] increase in the risk of NSCL/P, when compared to the A allele. Further, the rs560426 GG genotype showed 1.74- [28], 3.0- [29], 1.86- [30], and 1.72-fold [31] elevated risks of NSCL/P compared to the AA genotype. In contrast, the G allele (compared to the A allele), GG genotype [32] (compared to the AA genotype), AG genotype [7,32] (compared to the AA genotype), A allele [32] (compared to the C allele), AA genotype [32] (compared to the CC genotype), and the CA genotype [32] (compared to the CC genotype) had protective roles against NSCL/P. Further, when ethnicity was not considered, the present meta-analysis did not yield any significant risk of NSCL/P related to *ABCA4* polymorphisms (rs560426 and rs481931). In contrast, when ethnicity was considered, among Asians, both the G allele (compared to the A allele) and GG genotype (compared to AA genotype) were associated with a significantly increased risk of NSCL/P. Next, there is a need to further investigate the possible role of the *ABCA4* gene in the etiology of NSCL/P. This need stems from the following observations: First, some studies [30,31,32] indicated a link between the *ABCA4* gene and the risk of a NSCL/P; second, two GWAS studies [19,36] underscored the associations between the *ABCA4* gene with the risk of CL/P; third, the association between the *ABCA4* gene and the risk of NSCL/P was observed among at least three different populations, that is to say, among Taiwanese, Honduran and Columbian populations [20,35,37]. Next, Fontoura et al. [32] found significant associations between *ABCA4* rs481931 alleles and the status of bilateral and unilateral NSCL/P. Last, the replication of *ABCA4* polymorphisms among independent families from various populations revealed that Asian families showed evidence of a higher risk of NSCL/P related to polymorphisms of the *ABCA4* gene when compared to the European or American families [28]. Importantly, the present meta-analysis confirmed a stronger evidence of risk of NSCL/P related to the rs560426 polymorphism of the ABCA4 gene in Asian compared to other ethnicities.

One study [27] suggested that the strong correlation between the *ABCA4* gene and the risk of NSCL/P may be due to the impact of nearby genes (such as Rho GTPase Activating Protein 29 (*ARHGAP29*), and not due to the impact of a single locus. ARHGAP29 is expressed in the palate and lips of mice, its expression depends on Interferon Regulatory Factor 6 (*IRF6*), and coding variants in *ARHGAP29* are related to CL/P [12]. Therefore, evidence showed that *ARHGAP29* is the risk gene at 1p22 [38]. It follows that it is possible that the associated variants in *ABCA4* are in linkage disequilibrium with a causal variant included in a neighboring gene, and that associated variants in *ABCA4* act as indirect surrogates for a real etiologic variant in individuals with NSCL/P [32,33,37].

Besides a purely genetic-based explanation of the etiology of NSCL/P, both gene–gene and gene–environment interactions showed that there was an interaction between the *ABCA4* rs560426 polymorphism and folic acid consumption and the V-maf avian musculoaponeurotic fibrosarcoma oncogene homolog B rs11696257 polymorphism [7]. Further, the different frequencies of the G allele of rs560426 among different populations may identify the complex genetic etiology of NSCL/P and this polymorphism alone has no effect on the pathogenesis of the NSCL/P [30]. Transmission analysis using case-parent core pedigrees showed that the C allele of rs481931 was significantly over-transmitted from parents to children, indicating that the C allele is associated with the NSCL/P risk. In addition, the C (rs481931)-G (rs560426) haplotype was significantly involved in the occurrence of NSCL/P [8].

Last, a significantly different rate of the G allele of the rs560426 polymorphism among Asians, but not among Europeans, showed that the genetic variations can be associated with the pathogenesis of NSCL/P, but the role of ethnicity on genetic variations should be taken into consideration.

To summarize, the main findings of the present meta-analysis are as follows: the discrepancy between the results of GWAS studies, this meta-analysis, and individual studies suggest that *ABCA4* polymorphisms may be in an imbalance linked to polymorphism(s) located in other genes and *ABCA4* can be an indirect substitute for NSCL/P etiology. However, the novelty should be balanced against the following limitations. First, the number of studies was rather small. Second, samples were not further investigated for age, sex, and genotyping methods. Third, not all studies reported the Hardy–Weinberg equilibrium. Likewise, fourth, some sources of controls and ethnicities evaluated in the studies were different. However, sensitivity analysis identified the stability of the results, and also the heterogeneity across the studies was minimal.

## 5. Conclusions

The findings of the present meta-analysis confirmed that the G allele and GG genotype of the rs560426 polymorphism were significantly associated with the NSCL/P risk in the Asian population, while no such association was observed for the rs481931 polymorphism. There was no association between both polymorphisms (rs560426 and rs481931) and the risk of NSCL/P in the European descent and the mixed ethnicities. There were no differences in the risk of NSCL/P with rs560426 and rs481931 polymorphisms in relation to the source of the controls. Therefore, the ethnicity and the number of participants may act as significant factors in this association. It follows that further studies are needed, focusing on the gene–gene interactions of *ABCA4* and *ARHGAP29* with larger sample sizes in different ethnicities to understand their contribution to the present pattern of results.

## Figures and Tables

**Figure 1 life-11-00058-f001:**
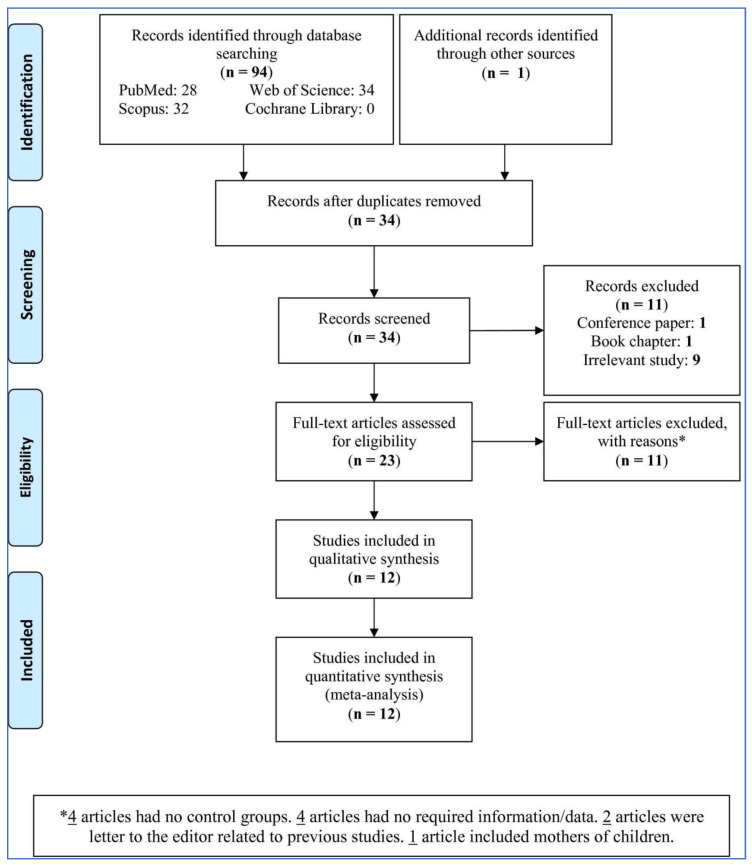
PRISMA flow-chart of the study selection.

**Figure 2 life-11-00058-f002:**
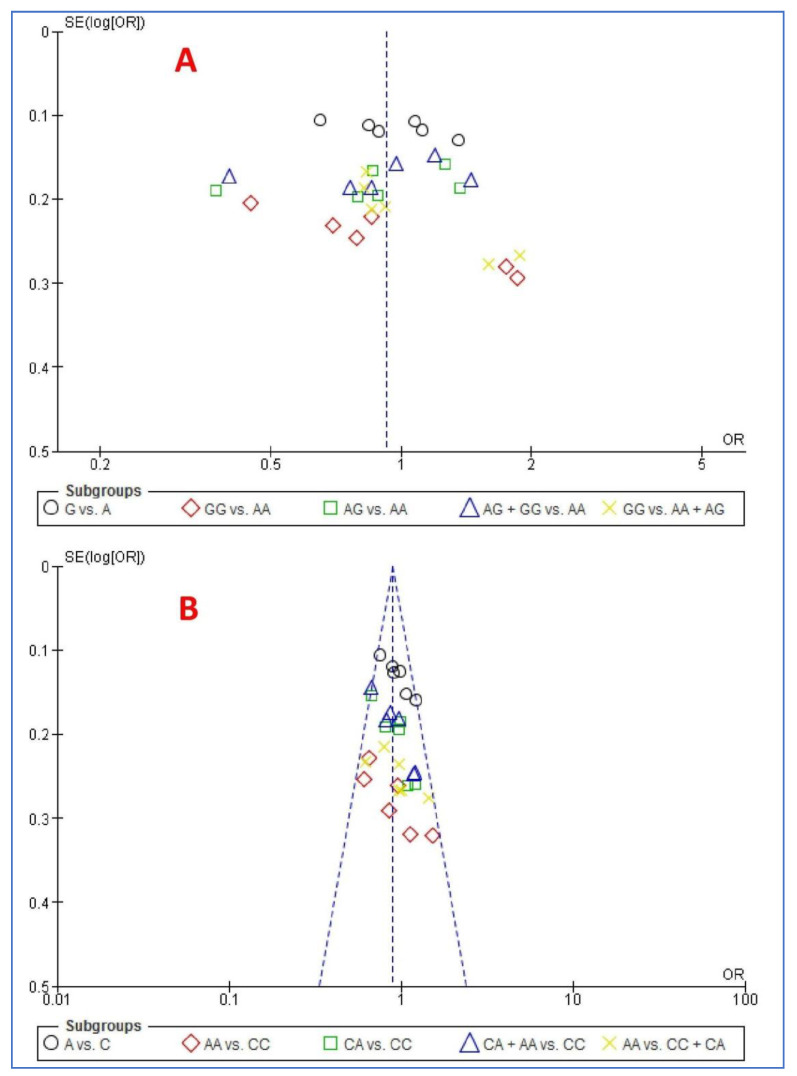
Funnel plot of the association between *ABCA4* polymorphisms and risk of NSCL/P using the five genetic models: (**A**) rs560426 and (**B**) rs481931.

**Table 1 life-11-00058-t001:** Characteristics of the studies included in this meta-analysis (*n* = 12).

First Author, (Year)	Ethnic Group	Source of Controls	Mean Age, Year (NSCL/P Patients to Controls)	No. of Males, (NSCL/P Patients to Controls)	ABCA4 rs560426		ABCA4 rs481931		Genotyping Method	*p*-Value for HWE in Controls
					AA/AG/GG		CC/CA/AA			
					Case	Control	Case	Control		
Pan et al. (2011) [27]	Asian	HB	5.54 to 5.49	246 to 242	145/175/51	167/160/57	NA	NA	TaqMan	0.071
Fontoura et al. (2012) [32]	European descent	HB	17.3 to 24.8	252 to 165	116/118/86	74/203/123	184/155/44	154/192/57	TaqMan	0.542/0.818
Huang et al. (2012) [28]	Asian	HB	NA	169 to 203	135/126/39	157/171/26	NA	NA	MALDI-TOF MS (Sequenom)	**0.024**
Mostowska et al. (2012) [33]	European descent	HB	NA	NA	62/105/39	120/230/96	79/98/29	156/196/94	PCR-HRM	0.467/**0.028**
Bagordakis et al. (2013) [31]	Mixed	PB	NA	NA	74/140/85	127/172/85	NA	NA	Multiplex PCR	0.067
Zhong-wei et al. (2013) [29]	Asian	PB	NA	NA	54/91/36	36/50/18	NA	NA	TaqMan	0.928
Ludwig et al. (2014) [9]	Mixed	PB	NA	102 to 111	37/73/33	100/163/66	NA	NA	MALDI-TOF MS (Sequenom)	0.977
do Rego Borges et al. (2015) [23]	Mixed	HB	NA	NA	76/152/65	74/187/91	NA	NA	TaqMan	0.223
Babu Gurramkonda et al. (2015) [34]	European descent	PB	NA	NA	46/72/26	61/80/35	41/68/35	57/87/32	**Kompetitive allele specific PCR (KASP)**	0.348/0.905
Mi et al. (2015) [30]	Asian	HB	4.98 to 5.20	NA	88/104/30	158/137/29	79/107/36	113/157/54	**Mini-sequencing (SNAPSHOT)**	0.928/0.965
Velázquez-Aragón et al. (2016) [7]	Mixed	PB	5.5 to 1.33	99 to 132	44/56/32	54/137/68	32/71/27	71/131/53	**Kompetitive allele specific PCR (KASP)**	0.326/0.602
Wu et al. (2018) [8]	Asian	PB	NA	NA	103/116/29	111/145/24	92/126/30	91/154/35	PCR-RFLP	**0.014/0.015**

**Abbreviations**: MALDI-TOF MS, Matrix-assisted laser desorption/ionization time-of-flight mass spectrometry; PCR, polymerase chain reaction; HRM, high resolution melting; RFLP, restriction fragment length polymorphism; NSCL/P, non-syndromic cleft lip/palate; HB, hospital-based; PB, population-based; HWE, Hardy–Weinberg equilibrium, NA, not available.

**Table 2 life-11-00058-t002:** Quality assessment scores for the studies included in this meta-analysis (*n* = 12).

First Author, (Year)	Selection (Four Points)	Comparability (Two Points)	Exposure (Three Points)	Total Points
Pan et al. (2011) [27]	***	**	***	8
Fontoura et al. (2012) [32]	***	-	***	7
Huang et al. (2012) [28]	***	*	***	7
Mostowska et al. (2012) [33]	***	**	***	8
Bagordakis et al. (2013) [31]	****	-	***	7
Zhong-wei et al. (2013) [29]	****	-	***	7
Ludwig et al. (2014) [9]	****	-	***	7
do Rego Borges et al. (2015) [23]	***	-	***	7
Babu Gurramkonda et al. (2015) [34]	****	**	***	9
Mi et al. (2015) [30]	***	**	***	8
Velázquez-Aragón et al. (2016) [7]	****	*	***	8
Wu et al. (2018) [8]	****	-	***	7

Each asterisk indicates one point. **Selection**: Is the case definition adequate? (one point), Representativeness of the cases (one point), Selection of Controls (one point), and Definition of Controls (one point). **Comparability:** Comparability of cases and controls on the basis of the design or analysis (two points). **Exposure**: Ascertainment of exposure (one point), Same method of ascertainment for cases and controls (one point), and Non-Response rate (one point).

**Table 3 life-11-00058-t003:** The results of Forest plot analysis of NSCL/P risk related to the *ABCA4* rs560426 polymorphism using the five genetic models.

Genetic Model	First Author, Publication Year	NSCL/P		Control		Weight	Odds Ratio
		Events	Total	Events	Total		M-H, Random, 95%CI
G vs. A	Pan, 2011	277	742	274	768	9.2%	1.07 [0.87, 1.32]
	Mostowska, 2012	183	412	422	892	8.7%	0.89 [0.70, 1.13]
	Huang, 2012	204	600	223	708	8.8%	1.12 [0.89, 1.41]
	Fontoura, 2012	290	640	449	800	9.2%	0.65 [0.53, 0.80]
	Zhong-wei, 2013	163	362	86	208	6.7%	1.16 [0.82, 1.64]
	Bagordakis, 2013	310	598	342	767	9.1%	1.34 [1.08, 1.66]
	Ludwig, 2014	139	286	295	658	7.9%	1.16 [0.88, 1.54]
	Mi, 2015	164	444	195	648	8.3%	1.36 [1.05, 1.76]
	do Rego Borges, 2015	282	586	369	704	9.0%	0.84 [0.68, 1.05]
	Babu Gurramkonda, 2015	124	288	150	352	7.2%	1.02 [0.74, 1.39]
	Velázquez-Aragón, 2016	120	264	273	518	7.5%	0.75 [0.56, 1.01]
	Wu, 2018	174	496	193	560	8.3%	1.03 [0.80, 1.32]
Subtotal (95%CI)			5718		7583	100.0%	1.01 [0.88, 1.15]
Total Events		2430		3271			
Heterogeneity: Tau^2^ = 0.04; Chi^2^ = 39.48, df = 11 (*p* < 0.0001); I^2^ = 72%; Test for overall effect: Z = 0.10 (*p* = 0.92)							
GG vs. AA	Pan, 2011	51	226	57	224	9.1%	0.85 [0.55, 1.32]
	Mostowska, 2012	39	101	96	216	8.8%	0.79 [0.49, 1.27]
	Huang, 2012	39	174	26	183	8.3%	1.74 [1.01, 3.01]
	Fontoura, 2012	86	202	123	197	9.4%	0.45 [0.30, 0.67]
	Zhong-wei, 2013	36	60	18	54	6.6%	3.00 [1.39, 6.45]
	Bagordakis, 2013	85	159	85	212	9.3%	1.72 [1.13, 2.60]
	Ludwig, 2014	33	70	66	166	8.1%	1.35 [0.77, 2.37]
	Mi, 2015	30	118	29	187	8.1%	1.86 [1.05, 3.29]
	do Rego Borges, 2015	65	141	91	165	9.0%	0.70 [0.44, 1.09]
	Babu Gurramkonda, 2015	26	72	35	96	7.6%	0.99 [0.52, 1.86]
	Velázquez-Aragón, 2016	32	76	68	122	8.0%	0.58 [0.32, 1.03]
	Wu, 2018	29	132	24	135	7.8%	1.30 [0.71, 2.38]
Subtotal (95%CI)			1531		1957	100.0%	1.08 [0.79, 1.47]
Total EVENTS		551		718			
Heterogeneity: Tau^2^ = 0.23; Chi^2^ = 47.66, df = 11 (*p* < 0.00001); I^2^ = 77%; Test for overall effect: Z = 0.47 (*p* = 0.64)							
AG vs. AA	Pan, 2011	175	320	160	327	9.3%	1.26 [0.92, 1.72]
	Mostowska, 2012	105	167	230	350	8.6%	0.88 [0.60, 1.30]
	Huang, 2012	126	261	171	328	9.2%	0.86 [0.62, 1.19]
	Fontoura, 2012	118	234	203	277	8.7%	0.37 [0.26, 0.54]
	Zhong-wei, 2013	91	145	50	86	6.9%	1.21 [0.70, 2.09]
	Bagordakis, 2013	140	214	172	299	8.8%	1.40 [0.97, 2.01]
	Ludwig, 2014	73	110	163	263	7.7%	1.21 [0.76, 1.93]
	Mi, 2015	104	192	137	295	8.8%	1.36 [0.95, 1.96]
	Babu Gurramkonda, 2015	72	118	80	141	7.4%	1.19 [0.73, 1.96]
	do Rego Borges, 2015	152	228	187	261	8.6%	0.79 [0.54, 1.16]
	Velázquez-Aragón, 2016	56	100	137	191	7.3%	0.50 [0.30, 0.83]
	Wu, 2018	116	219	145	256	8.8%	0.86 [0.60, 1.24]
Subtotal (95% CI)			2308		3074	100.0%	0.93 [0.73, 1.17]
Total Events		1328		1835			
Heterogeneity: Tau^2^ = 0.13; Chi^2^ = 46.53, df = 11 (*p* < 0.00001); I^2^ = 76%; Test for overall effect: Z = 0.63 (*p* = 0.53)							
AG + GG vs. AA	Pan, 2011	226	371	217	384	9.1%	1.20 [0.90, 1.60]
	Fontoura, 2012	204	320	326	400	8.7%	0.40 [0.28, 0.56]
	Huang, 2012	165	300	197	354	8.9%	0.97 [0.71, 1.33]
	Mostowska, 2012	144	206	326	446	8.5%	0.85 [0.59, 1.23]
	Zhong-wei, 2013	127	181	68	104	7.2%	1.25 [0.74, 2.08]
	Bagordakis, 2013	225	299	257	384	8.7%	1.50 [1.07, 2.11]
	Ludwig, 2014	106	143	229	329	7.8%	1.25 [0.80, 1.95]
	Mi, 2015	134	222	166	324	8.6%	1.45 [1.03, 2.05]
	do Rego Borges, 2015	217	293	278	352	8.5%	0.76 [0.53, 1.10]
	Babu Gurramkonda, 2015	68	144	115	176	7.7%	0.47 [0.30, 0.75]
	Velázquez-Aragón, 2016	88	132	205	259	7.6%	0.53 [0.33, 0.84]
	Wu, 2018	145	248	169	280	8.6%	0.92 [0.65, 1.31]
Subtotal (95%CI)			2859		3792	100.0%	0.89 [0.70, 1.14]
Total Events		1849		2553			
Heterogeneity: Tau^2^ = 0.15; Chi^2^ = 59.28, df = 11 (*p* < 0.00001); I^2^ = 81%; Test for overall effect: Z = 0.90 (*p* = 0.37)							
GG vs. AA + AG	Pan, 2011	51	371	57	384	9.6%	0.91 [0.61, 1.37]
	Fontoura, 2012	86	320	123	400	12.2%	0.83 [0.60, 1.15]
	Huang, 2012	39	300	26	354	6.9%	1.89 [1.12, 3.18]
	Mostowska, 2012	39	206	96	446	9.4%	0.85 [0.56, 1.29]
	Bagordakis, 2013	85	299	85	384	11.4%	1.40 [0.99, 1.98]
	Zhong-wei, 2013	36	181	18	104	5.3%	1.19 [0.63, 2.22]
	Ludwig, 2014	33	143	66	329	7.9%	1.20 [0.74, 1.92]
	do Rego Borges, 2015	65	293	91	352	10.9%	0.82 [0.57, 1.18]
	Mi, 2015	30	222	29	324	6.6%	1.59 [0.92, 2.73]
	Babu Gurramkonda, 2015	26	144	35	176	6.2%	0.89 [0.51, 1.56]
	Velázquez-Aragón, 2016	32	132	68	259	7.7%	0.90 [0.55, 1.46]
	Wu, 2018	29	248	24	280	6.1%	1.41 [0.80, 2.50]
Subtotal (95%CI)			2859		3792	100.0%	1.08 [0.91, 1.26]
Total Events		551		718			
Heterogeneity: Tau^2^ = 0.03; Chi^2^ = 17.14, df = 11 (*p* = 0.10); I^2^ = 36%; Test for overall effect: Z = 0.87 (*p* = 0.38)							

**Abbreviations**: NSCL/P, non-syndromic cleft lip with or without a cleft palate; CI, confidence interval. All models were analyzed based on random-effects model except for “GG vs. AA + AG” that was based on fixed-effects model

**Table 4 life-11-00058-t004:** The results of Forest plot analysis of NSCL/P risk related to the *ABCA4* rs481931 polymorphism using the five genetic models.

Genetic Model	First Author, Publication Year	NSCL/P		Control		Weight	Odds Ratio
		Events	Total	Events	Total		M-H, Fixed, 95%CI
A vs. C	Fontoura, 2012	243	766	306	806	26.6%	0.76 [0.62, 0.94]
	Mostowska, 2012	183	412	422	892	19.4%	0.89 [0.70, 1.13]
	Mi, 2015	179	444	265	648	16.8%	0.98 [0.76, 1.25]
	Babu Gurramkonda, 2015	138	288	151	352	9.2%	1.22 [0.90, 1.67]
	Velázquez-Aragón, 2016	125	260	237	510	10.9%	1.07 [0.79, 1.44]
	Wu, 2018	186	496	224	560	17.2%	0.90 [0.70, 1.15]
Subtotal (95%CI)			2666		3768	100.0%	0.92 [0.83, 1.02]
Total Events		1054		1605			
Heterogeneity: Chi^2^ = 7.74, df = 5 (*p* = 0.17); I^2^ = 35%; Test for overall effect: Z = 1.57 (*p* = 0.12)							
AA vs. CC	Fontoura, 2012	44	228	57	211	26.6%	0.65 [0.41, 1.01]
	Mostowska, 2012	29	108	94	250	23.1%	0.61 [0.37, 1.00]
	Babu Gurramkonda, 2015	35	76	32	89	8.8%	1.52 [0.81, 2.84]
	Mi, 2015	36	115	54	167	16.8%	0.95 [0.57, 1.59]
	Velázquez-Aragón, 2016	27	59	53	124	10.3%	1.13 [0.61, 2.11]
	Wu, 2018	30	122	35	126	14.4%	0.85 [0.48, 1.50]
Subtotal (95%CI)			708		967	100.0%	0.85 [0.68, 1.05]
Total Events		201		325			
Heterogeneity: Chi^2^ = 7.49, df = 5 (*p* = 0.19); I^2^ = 33%; Test for overall effect: Z = 1.52 (*p* = 0.13)							
CA vs. CC	Fontoura, 2012	155	339	192	346	31.0%	0.68 [0.50, 0.91]
	Mostowska, 2012	98	177	196	352	17.6%	0.99 [0.69, 1.42]
	Mi, 2015	107	186	157	270	16.3%	0.97 [0.67, 1.42]
	Babu Gurramkonda, 2015	68	109	87	144	8.5%	1.09 [0.65, 1.81]
	Velázquez-Aragón, 2016	71	103	131	202	8.3%	1.20 [0.72, 2.00]
	Wu, 2018	126	218	154	245	18.4%	0.81 [0.56, 1.18]
Subtotal (95% CI)			1132		1559	100.0%	0.88 [0.75, 1.03]
Total Events		625		917			
Heterogeneity: Chi^2^ = 5.93, df = 5 (*p* = 0.31); I^2^ = 16%; Test for overall effect: Z = 1.57 (*p* = 0.12)							
CA + AA vs. CC	Fontoura, 2012	199	383	249	403	31.1%	0.67 [0.50, 0.89]
	Mostowska, 2012	127	206	290	446	18.7%	0.86 [0.61, 1.22]
	Babu Gurramkonda, 2015	103	144	119	176	8.1%	1.20 [0.74, 1.95]
	Mi, 2015	143	222	211	324	16.3%	0.97 [0.68, 1.39]
	Velázquez-Aragón, 2016	98	130	184	255	8.2%	1.18 [0.73, 1.92]
	Wu, 2018	156	248	189	280	17.6%	0.82 [0.57, 1.17]
Subtotal (95%CI)			1333		1884	100.0%	0.87 [0.75, 1.00]
Total Events		826		1242			
Heterogeneity: Chi^2^ = 7.05, df = 5 (*p* = 0.22); I^2^ = 29%; Test for overall effect: Z = 1.91 (*p* = 0.06)							
AA vs. CC + CA	Mostowska, 2012	29	206	94	446	23.6%	0.61 [0.39, 0.97]
	Fontoura, 2012	44	383	57	403	22.8%	0.79 [0.52, 1.20]
	Babu Gurramkonda, 2015	35	144	32	176	10.1%	1.44 [0.84, 2.48]
	Mi, 2015	36	222	54	324	17.0%	0.97 [0.61, 1.53]
	Velázquez-Aragón, 2016	27	130	53	255	13.1%	1.00 [0.59, 1.68]
	Wu, 2018	30	248	35	280	13.4%	0.96 [0.57, 1.62]
Subtotal (95%CI)			1333		1884	100.0%	0.89 [0.74, 1.09]
Total Events		201		325			
Heterogeneity: Chi^2^ = 6.39, df = 5 (*p* = 0.27); I^2^ = 22%; Test for overall effect: Z = 1.12 (*p* = 0.26)							

**Abbreviations**: NSCL/P, non-syndromic cleft lip with or without a cleft palate; CI, confidence interval.

**Table 5 life-11-00058-t005:** Analysis of non-syndromic cleft lip/palate risk related to the rs560426 polymorphism according to ethnicity and source of controls.

Variable (N)	G vs. A	GG vs. AA	AG vs. AA	AG + GG vs. AA	GG vs. AA + AG
	OR (95%CI), I^2^ (%), *P*_h_	OR (95%CI), I^2^ (%), *P*_h_	OR (95%CI), I^2^ (%), *P*_h_	OR (95%CI), I^2^ (%), *P*_h_	OR (95%CI), I^2^ (%), *P*_h_
Overall (12)	1.01 (0.88, 1.15), 72, <0.0001	1.08 (0.79, 1.47), 77, <0.00001	0.93 (0.73, 1.16), 76, <0.00001	0.89 (0.70, 1.14), 81, <0.00001	1.08 (0.91, 1.26), 36, 0.10
Ethnicity					
Asian (5)	**1.13 (1.01, 1.27), 0, 0.59**	**1.53 (1.01, 2.31), 62, 0.03**	1.08 (0.92, 1.26), 35, 0.19	1.13 (0.97, 1.32), 10, 0.35	**1.30 (1.03, 1.63), 27, 0.24**
European Descent (3)	0.82 (0.63, 1.08), 71, 0.03	0.67 (0.42, 1.09), 64, 0.06	0.72 (0.36, 1.45), 88, 0.0002	0.55 (0.34, 0.88), 79, 0.009	0.85 (0.67, 1.07), 0, 0.98
Mixed (4)	1.00 (0.76, 1.31), 79, 0.003	0.99 (0.59, 1.68), 78, 0.004	0.92 (0.60, 1.42), 76, 0.006	0.94 (0.60, 1.49), 81, 0.001	1.07 (0.87, 1.30), 41, 0.17
Source of Controls					
Hospital-Based (6)	0.94 (0.86, 1.03), 80, 0.0001	0.83 (0.69, 1.01), 80, 0.0002	0.87 (0.76, 1.01), 84, <0.00001	0.89 (0.78, 1.02), 85, <0.00001	0.98 (0.83, 1.15), 57, 0.04
Population-Based (6)	1.07 (0.91, 1.26), 52, 0.06	1.29 (0.86, 1.94), 66, 0.01	1.02 (0.76, 1.36), 60, 0.03	0.91 (0.62, 1.33), 80, 0.0002	1.18 (0.97, 1.44), 0, 0.63

Bold numbers mean statistically significant (*p* < 0.05). *P*_h_ indicates *P*_heterogeneity_. Abbreviations: OR, odds ratio; CI, confidence interval. “N” shows the number of studies.

**Table 6 life-11-00058-t006:** Analysis of non-syndromic cleft lip/palate risk related to the rs481931 polymorphism according to ethnicity and source of controls.

Variable (N)	A vs. C	AA vs. CC	CA vs. CC	CA + AA vs. CC	AA vs. CC + CA
	OR (95%CI), I^2^ (%), *P*_h_	OR (95%CI), I^2^ (%), *P*_h_	OR (95%CI), I^2^ (%), *P*_h_	OR (95%CI), I^2^ (%), *P*_h_	OR (95%CI), I^2^ (%), *P*_h_
Overall (6)	0.92 (0.83, 1.02), 35, 0.17	0.85 (0.68, 1.05), 33, 0.19	0.88 (0.75, 1.03), 16, 0.31	0.87 (0.75, 1.00), 0.29, 0.22	0.89 (0.74, 1.09), 22, 0.27
Ethnicity					
Asian (2)	0.94 (0.79, 1.12), 0, 0.65	0.90 (0.62, 1.32), 0, 0.76	0.89 (0.68, 1.16), 0, 0.49	0.89 (0.69, 1.15), 0, 0.51	0.97 (0.68, 1.36), 0, 0.99
European Descent (3)	0.92 (0.71, 1.18), 68, 0.04	0.81 (0.48, 1.36), 67, 0.05	0.83 (0.67, 1.03), 46, 0.15	0.85 (0.62, 1.16), 56, 0.11	0.87 (0.55, 1.38), 66, 0.05
Mixed (1)	1.07 (0.79, 1.44)	1.13 (0.61, 2.11)	1.20 (0.72, 2.00)	1.18 (0.73, 1.92)	1.00 (0.59, 1.68)
Source of Controls					
Hospital-Based (2)	0.85 (0.67, 1.09), 57, 0.13	0.77 (0.55, 1.07), 21, 0.26	0.80 (0.56, 1.14), 55, 0.14	0.79 (0.55, 1.14), 61, 0.11	0.86 (0.63, 1.18), 0, 0.52
Population-Based (4)	0.98 (0.86, 1.12), 11, 0.34	0.91 (0.68, 1.20), 47, 0.13	0.97, (0.79, 1.20), 0, 0.62	0.95 (0.78, 1.16), 0, 0.44	0.91 (0.71, 1.17), 49, 0.12

*p*-Value > 0.05 in all analyses and *P*_h_ indicates *P*_heterogeneity_. Abbreviations: OR, odds ratio; CI, confidence interval. “N” shows the number of studies.

**Table 7 life-11-00058-t007:** Meta-regression analysis based on publication year for association between rs560426 and rs481931 polymorphisms and the risk of non-syndromic cleft lip/palate.

Variable	Polymorphism		Allele	Homozygote	Heterozygote	Recessive	Dominant
**Publication Year**	rs560426	R	0.025	0.047	0.113	0.172	0.075
		Adjusted R^2^	−0.099	−0.098	−0.086	−0.068	−0.094
		*P*	0.937	0.884	0.726	0.593	0.816
	rs481931	R	0.406	0.456	0.243	0.388	0.495
		Adjusted R^2^	−0.044	0.010	−0.176	−0.062	0.057
		*P*	0.424	0.364	0.643	0.447	0.318
**Number of Participants**	rs560426	R	0.098	0.410	0.171	0.118	0.036
		Adjusted R^2^	−0.089	0.085	−0.068	−0.085	−0.099
		*P*	0.761	0.185	0.594	0.714	0.912
	rs481931	R	**0.953**	**0.913**	0.810	0.650	**0.814**
		Adjusted R^2^	**0.886**	**0.793**	0.570	0.279	**0.579**
		*P*	**0.003**	**0.011**	0.051	0.162	**0.049**

R: Correlation coefficient. Bold numbers mean statistically significant (*p* < 0.05).

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
