# Peer review of "Polymorphisms of ATP-Binding Cassette, Sub-Family A, Member 4 (rs560426 and rs481931) and Non-Syndromic Cleft Lip/Palate: A Meta-Analysis"

_life, 2021, doi:10.3390/life11010058_

Round 1

Reviewer 1 Report

Imani and colleagues performed a meta-analysis of genotype data of two SNPs (rs560426 and rs481931) located in intronic regions of ABCA4 to assess their contribution to nsCLP risk. One major issue is that both polymorphisms (rs560426 and rs481931) already reached genome-wide significance in GWASs: e.g. rs560426 (3.14x10^-12) in Beaty et al. 2010 in an admixed sample (European and Asian) and rs481931 (1.06x10^-12) in a study by Yu and colleagues (2017) in an Asian cohort, also in different study populations. Therefore, the aim of the study is somehow questionable to me.

Furthermore, the authors claim that both SNPs are polymorphisms of the gene ABCA4. However, both are located within intronic (noncoding) regions and might have also regulatory effect on other genes at the locus. Already Beaty and colleagues (2010) had doubts about ABCA4 being the etiologic gene at the 1p22 locus. This was further assessed in a study by Leslie et al. 2012, who found ARHGAP29 to be most likely the etiologic gene at the locus. This essential fact is not described or referenced in the introduction and just mentioned briefly in the discussion and the authors even used ABCA4 in the title of the paper what is somehow misleading to me.

In general, I miss essential literature on nsCL/P risk loci (at least the largest GWAS for this trait) and the use of the GWAS catalogue. This tool would have found the genome-wide significant associations of both polymorphisms (Ludwig et al. 2012 and Yu et al 2017; both found with GWAS catalogue). Except for the Beaty et al. 2010, I do not find other studies that found genome-wide significant associations for the two polymorphisms in the references.

Further points:

21: It is 40 loci.

23: Are these really polymorphisms in ABCA4? Or are these located near ABCA4? To my knowledge both are located within intronic (noncoding) regions and might have also regulatory effect on other genes at the locus. Already Beaty and colleagues (2010) had doubts about ABCA4 being the etiologic gene at the 1p22 locus. This was further assessed in a study by Leslie et al. 2012, who found ARHGAP29 to be most likely the etiologic gene at the locus. This essential fact is not described or referenced in the introduction and just mentioned briefly in the discussion and the authors even used ABCA4 in the title of the paper what is somehow misleading to me.

24: Why not using GWAS catalogue?

27-28: Why not genotyping method?

38: rs560426 has reached genome-wide significance (3.14x10^-12) in Beaty et al. 2010 in an admixed sample (European and Asian) already. Also, rs481931 reached genome-wide significance (1.06x10^-12) in a study by Yu and colleagues (2017) in an Asian cohort.

44: This should read ‘70% of all CL/Ps present without further anomalies and are thus nonsyndromic’.

46: Study outcomes concerning environmental factors are contradictory, except for maternal smoking. This should be mentioned.

50: ‘Subsequent genome-wide association studies showed that a number of gene loci are closely associated with the incidence of NSCL/P [7,8].’ References 7 and 8 are not sufficient, given that there are 40 loci known to date. What does the catalogue mean here?

60-61: To date, there are more than 40 known nsCL/P risk loci, most of them identified by GWAS and follow-up analyses, that the authors do not report on in the introduction. Then, based on contradictory results of two polymorphisms they claim that the genetic basis of nsCL/P is unclear. This is not the complete picture.

69: GWAS catalogue is missing.

133-134: The term ABCA4 polymorphism is misleading. It´s intronic variants that may have a regulatory effect also on other genes.

201-203: I do not understand the sentence.

232: ‘in that associations of specific genetic polymorphisms the predict the risk of NSCL/P was observed among the Asian, but not among the Caucasian ethnicity.’ To my knowledge, this has been shown in Ludwig et al. 2012 (first hit in GWAS catalogue when searching for rs560426) where the SNP reached genome-wide significance in a meta-analysis after adding the data of the Asian study population.

235-236: As nsCL/P is a trait with a complex genetic background this statement is overrating the outcome of the study.

241: See ARHGAP29 issue.

262: ABCA4 gene presence?

265: Linkage disequilibrium.

280-281: “Given this, it appears rather unlikely that the genetic polymorphism is the sole cause of NSCL/P.” Given that the authors are writing about genetic association data, which is based on common variants with low effect sizes this is nothing that is new but rather essential knowledge in the field of complex genetics.

Author Response

Thank you for your kind efforts. Please find the detailed point-by-point response attached as a separate file. 

Reviewer 2 Report

In this study, the authors sought to characterize the relationship between two SNPs (rs560426 and rs481931) in ABCA4 and the odds of non-syndromic cleft lip with or without cleft palate (NSCL/P). They used meta-analysis of 12 studies with adequate available data for these SNPs and NSCL/P to estimate the odds ratios.  Overall, they did not observe a significant association between either SNP and odds of NSCL/P; however, in the Asian ancestry subgroup analysis, a significant association was seen between rs560426 and odds of NSCL/P. The observed associations are quite modest, a stark contrast to existing literature that demonstrates a strong statistical association signal near ABCA4. The methods are mostly adequate; however, without a broader scope or better contextualization of findings, the conclusions of this work are quite limited.

Major comments:

  • The choice to examine only 2 SNPs in ABCA4 is not fully justified. The SNPs are in quite low LD with each other (0 < R2 < 0.17 in 1000G populations) but the association signal at the 1p22 locus is very broad. This represents a severe limitation of this study: it assumes that these two variants follow similar LD patterns across ancestries that they would mark the association of the potential causal variant(s) near ABCA4.
  • Please use “European descent” (or “European ancestry”, etc.) in place of “Caucasian”. The term “Caucasian” is not typically used in modern genetic and epidemiologic studies because of its racist origins.
  • More details are needed in the methods and results for the sensitivity analysis, including what “stable and trustworthy” results mean quantitatively. For which methods was a sensitivity analysis conducted? How were the results evaluated?
  • There are several problems with the conclusions presented:
    • The implication that this information should be used in clinical genetic counseling settings (lines 235-236) is unwarranted based on the results presented here.
    • There have been other studies that attempt to fine-map the association signal in this region and suggest that the pathophysiology underlying the GWAS signal near ABCA4 is actually due to causal variation in ARHGAP29 – there are known associations between coding variation in ARHGAP29 and NSCL/P risk and ARHGAP29 expression depends on IRF6, and another well-studied NSCL/P gene.[1] The results of this meta-analysis should be contextualized within this framework.
    • As the authors point out in the introduction, NSCL/P is a complex birth defect with many associated genetic and environmental risk factors. The insinuation that one SNP could possibly explain the pathogenesis of NSCL/P (lines 278-281), given the current evidence of the genetic and environmental etiology of NSCL/P, is unfounded.
    • It is unclear what is meant by the novelty of these results (lines 282-285). This region, including these variants, have been reported previously. Moreover, ABCA4 is only one of several NSCL/P related genes – none of which can uniquely determine NSCL/P etiology.

Minor comments:

  • Was there any overlap in the participants in the 12 studies used?
  • Which ancestries comprised “Mixed” ancestry?
  • Why was an additive genetic model not tested additionally? This is a commonly used model in genetic associations.
  • It would be helpful to have allele frequencies (AF) presented in Table 2. How to these compare to global estimates of the AF for these two variants?
  • The conclusions presented in lines 160 & 186 to reflect that no significant association was observed, not that
  • The term “pooled analysis” is misleading. It typically refers to one in which all data are aggregated, in contrast to meta-analysis which combines effects across groups, weighting by sample size or standard error.
  • Formatting/typographic issues:
    • All instances of human gene names should be italicized
    • “association significant” (lines 58, 60) should be “significant association”
    • Table #s should be sequential (Table 4 appears before Tables 2&3)
    • Order of studies in Figures 2 and 3 should match that in Table 1
    • Use the label Ph consistently throughout
    • The following sentences could use rewording: lines 50, 134-136, 154-158, 244-248
    • Line 265: “connection disequilibrium” should be “linkage disequilibrium”

[1] Liu, H., Leslie, E., Carlson, J. et al. Identification of common non-coding variants at 1p22 that are functional for non-syndromic orofacial clefting. Nat Commun 8, 14759 (2017). https://doi.org/10.1038/ncomms14759

Author Response

Thank you for your kind efforts. Please find the detailed point-by-point-response attached as a separate file.

Reviewer 3 Report

In the manuscript entitled: “Polymorphisms of ATP-binding cassette, sub-family A, member 4 (rs560426 and rs481931) and non- syndromic cleft lip/palate: A meta-analysis” the authors evaluated the association between ATP-binding cassette, sub-family A, member 4 (ABCA4) polymorphisms (rs560426 and rs481931) and the NSCL/P risk by reviewing case-control studies.

The auhtors found that there was no significant association between the both polymorphisms and the risk of NSCL/P. However, subgroup analysis demonstrated that there was a higher risk of NSCL/P for specific models: the allelic model (OR = 1.13; p = 0.03), the homozygote model (OR = 1.53; p = 0.04), and the recessive model (OR = 1.30; p = 0.03) in the Asian ethnicity for rs560426 polymorphism.

The authors concluded that the NSCL/P risk was significantly associated with the G allele and GG genotype of rs560426 polymorphism but not for rs481931 polymorphism. There were no associations between the both polymorphisms (rs560426 and rs481931) and the NSCL/P risk in the Caucasian and the mixed ethnicities

Major comments:

In general, the idea and innovation of this meta-analysis, regards the analysis of polymorphisms of ATP-binding cassette in non- syndromic cleft lip/palate is interesting, because the analysis of these adjuvants is validated but further studies on this topic could be an innovative issue in this field could be open an innovative matter of debate in literature by adding new information. Moreover, there are few reports in the literature that studied this interesting topic with this kind of study design.

The study was well conducted by the authors; However, there are some concerns to revise that are described below.

The introduction section resumes the existing knowledge regarding the important factor linked with cleft lip/palate.

However, as the importance of the topic, the reviewer strongly recommends, before a further re-evaluation of the manuscript, to update the literature through read, discuss and must cites in the references with great attention all of those recent interesting articles, that helps the authors to better introduce and discuss the aim of the study in light of some others adjuvants and mediators of periodontitits: 1) Perillo L, Isola G, Esercizio D, Iovane M, Triolo G, Matarese G. Differences in craniofacial characteristics in Southern Italian children from Naples: a retrospective study by cephalometric analysis. Eur J Paediatr Dent. 2013 Sep;14(3):195-8.  2) Leonardi RM, Aboulazm K, Giudice AL, Ronsivalle V, D'Antò V, Lagravère M, Isola G. Evaluation of mandibular changes after rapid maxillary expansion: a CBCT study in youngsters with unilateral posterior crossbite using a surface-to-surface matching technique. Clin Oral Investig. 2020 Aug 2. doi: 10.1007/s00784-020-03480-5.

The authors should be better specified, at the end of the introduction section, the rational of the study and the aim of the meta-analysis. In the material and methods section, should better clarify the exac period of screening, the numer of clinicians involved and the risk of bias.

The discussion section appears well organized with the relevant paper that support the conclusions, even if the authors should better discuss the importance of malocclusions involved in cleft palate patients. The conclusion should reinforce in light of the discussions.

In conclusion, I am sure that the authors are fine clinicians who achieve very nice results with their adopted protocol. However, this study, in my view, does not in its current form, satisfy a very high scientific requirement for publication in this journal and requests a revision before a further re-evaluation of the manuscript.

Minor Comments:

Abstract:

  • Better formulate the introduction section by better describing the background

Introduction:

  • Please refer to major comments

Discussion

  • Please add a specific sentence that clarifies the results obtained in the first part of the discussion
  • Page 12 last paragraph: Please reorganize this paragraph that is not clear

Author Response

(The authors gave the same response as above.)

Round 2

Reviewer 2 Report

The authors have responded to many of the reviewers’ comments adequately. I have no further comments at this time.  

Reviewer 3 Report

The authors have well addressed to all reviewers comments. I suggest the acceptance of this interesting manuscript.

This manuscript is a resubmission of an earlier submission. The following is a list of the peer review reports and author responses from that submission.

Round 1

Reviewer 1 Report

1) In the Results section, from 34 studies selected after duplicated exclusion, the authors selected 23 and the reason why 11 were excluded must be included also in the flow chart. In addition, the authors mentioned that some studied showed no HWE, but these studies were not excluded and the reason why is not explained. The NOS score must be presented previous to the results of meta-analysis. Funnel plot figure does not show the funnel in panel A. 

2) In the discussion section the authors must comment about the mean of each genetic model in the risk associated to the phenotype, i.e. the mean of additive, recessive and dominat model related to the risk of present teh phenotype. 

3) Regarding the phrase "The quality score of all studies was high (≥ 7) and therefore 216 we couldn’t use a sensitivity analysis (excluding the studies with low quality) for this topic.", I think that the low quality studies must be excluded before the meta-analysis which is related with the point 1 of my review. 

4) Line 237 says "showed that the number of participants led to xxxxx". What is that?

Reviewer 2 Report

First, I would like to thank the authors of the study for their effort to prepare this revised version of the manuscript. However, I still miss (as mentioned in my first revision) essential literature on nsCL/P risk loci and the use of the GWAS catalogue. One important point that was not obvious from the first version of the manuscript was that the authors excluded studies without ‘full texts’. In my point of view, doing so in such a systematic analysis introduces bias and is thus problematic. Furthermore, the aim of the study to ‘assess the possible role of these polymorphisms in the etiology of NSCL/P (line 66-67 in the Abstract)’ is still questionable, since GWAS and meta-analyses in nsCL/P already showed a genome-wide significant association of both polymorphisms (Ludwig et al. 2012 and Yu et al 2017; both found with GWAS catalogue).

Besides these major points some comments are listed in the following:

line 45: Should this read ‘70% of all nsCL/Ps present without further anomalies and are thus nonsyndromic’?

line 49: Study outcomes concerning environmental factors are contradictory, except for maternal smoking. This should be mentioned.

line 52/53: ‘Subsequent genome-wide association studies showed that a number of gene loci are closely associated with the incidence of NSCL/P [7,8].’ This revised version still lacks essential literature on (40!) nsCL/P risk loci. References 7 and 8 are not sufficient.

line 62-63: To date, there are more than 40 known nsCL/P risk loci, most of them identified by GWAS and follow-up analyses, that the authors do not report in the introduction. Then, based on contradictory results of two polymorphisms they claim that the genetic basis of nsCL/P is unclear. This is not the complete picture.

line 72: Still, I miss the GWAS catalogue in this revised version of the manuscript.

line 118: Excluding studies without ‘full texts’ from my point of view is no scientific reason to exclude a study. This might introduce a bias and you might have excluded essential studies.

line 196: As mentioned by Reviewer #1, Table 4 giving the NOS scores of the individual studies should be shown BEFORE presenting the results. This is still not the case in revised version of the manuscript.

Line 199-200: Why not genotyping method?

line 232: This is in line with recent GWAS studies (see GWAS catalogue). Should be mentioned here.

line 235-236: ‘in that associations of specific genetic polymorphisms the predict the risk of NSCL/P was observed among the Asian, but not among the Caucasian ethnicity.’ To my knowledge, this has been shown in Ludwig et al. 2012 (first hit in GWAS catalogue when searching for rs560426) where the SNP reached genome-wide significance in a meta-analysis after adding the results of the Asian study population).

line 237: ‘the meta-regression showed that the number of participants led to xxxxx.’ What does the ‘xxxxx’ mean?

Line 284-285: “Given this, it appears rather unlikely that the genetic polymorphism is the sole cause of NSCL/P.” Given that, the authors are writing about genetic association data, which is based on common variants with low (to moderate) effect sizes this is nothing that is new but rather essential knowledge in the field of complex genetics (association does not equal causation).